# Partial Smoking Ban and Secondhand Smoke Exposure in Japan

**DOI:** 10.3390/ijerph16152804

**Published:** 2019-08-06

**Authors:** Sen Zeng, Haruko Noguchi, Satoru Shimokawa

**Affiliations:** 1Graduate School of Economics, Waseda University, 1-6-1 Nishi-Waseda, Shinjuku-ku, Tokyo 169-8050, Japan; 2Faculty of Political Science and Economics, Waseda University, 1-6-1 Nishi-Waseda, Shinjuku-ku, Tokyo 169-8050, Japan

**Keywords:** partial smoking ban, secondhand smoke exposure, difference-in-differences, Japan

## Abstract

Implementing smoking bans is a worldwide common practice for tobacco control. However, if the policy prohibits smoking partially rather than comprehensively, it may increase nonsmokers’ exposure to secondhand smoke (SHS) in nonprohibited places. This paper investigates how a partial smoking ban affected nonsmokers’ SHS exposure (measured by frequency of having exposure to SHS in days per month) in households, workplaces, and restaurants by examining the case of a partial smoking ban introduced in a large Japanese prefecture in 2013. Using data from the National Health and Nutrition Survey (NHNS) in 2010, 2013, and 2016 (n = 30,244) and the Comprehensive Survey of Living Conditions (CSLC) from 2001 to 2016 (n = 2,366,896), this paper employs a difference-in-differences (DID) approach. We found that the partial smoking ban significantly increased their SHS exposure in households and workplaces by 2.64 days and 4.70 days per month, respectively, while it did not change nonsmokers’ SHS exposure in restaurants. The results imply that the smoking ban displaced smokers from public places to private places. We also found that neither smokers’ smoking status nor smoking intensity changed significantly after implementing the partial smoking ban. Comprehensive smoking bans are needed to better protect nonsmokers from SHS exposure.

## 1. Introduction

Exposure to secondhand smoke (SHS) may cause serious illnesses, such as lung cancer, heart disease, and respiratory disease. SHS was estimated to account for 603,000 deaths and 10.9 million disability-adjusted life years (DALYs) lost in 2004 [1]. Thus, to prevent exposure to SHS, many countries have implemented the legislation of smoking bans in public places (e.g., Ireland, England, New Zealand, Netherlands, Malaysia, and Korea). However, the effectiveness of such policies depends on the comprehensiveness of their design. A partial smoking ban (smoking ban with exemptions or designated smoking rooms, e.g., under the partial smoking in Japan, restaurants, bars, and leisure facilities could choose either to prohibit smoking or introduce smoking separation, some small restaurants were even exempted from the ban) might have adverse impacts because smokers can change their smoking locations from public places (e.g., transport facilities, restaurants, bars) to private places (e.g., homes) without curbing their tobacco consumption.

Several studies have examined how legislative smoking bans influence SHS exposure in public places. A systematic review showed that evidences from interrupted time-series designs confirmed reductions in preterm birth and hospital attendances for asthma following smoking bans [2]. In other studies, smoking bans have been found to be associated with reduced SHS exposure and improved health [3,4,5,6,7], and they have also improved smoking behaviors in workplaces [8,9] and in bars and restaurants [10]. Some studies also found that cotinine (a metabolite of nicotine) concentrations decreased among hospitality workers [11,12,13], children [14], and nonsmoking pregnant women [15] after the implementation of indoor public smoking bans. In contrast, two studies revealed that smoking bans increased nonsmokers’ exposure to SHS due to a displacement of smokers from public places to private places (homes) [16,17]. However, to the best of our knowledge, no study has examined how a partial smoking ban with exemptions for the workplace and many restaurants influences nonsmokers’ SHS exposure at different locations in the context of Japan.

Among developed countries, Japan is often referred to as a smoker’s paradise and ranks among the least protected countries by the World Health Organization (WHO) because it does not have any binding laws controlling SHS. SHS exposure is estimated to claim 15,000 lives in Japan annually. Although Japan became a party to the WHO Framework Convention on Tobacco Control on February 27 in 2005, tobacco control policies are still weak in Japan. Municipal regulation of street smoking bans is a common practice nationwide, while the health impact of exposure to SHS is not clearly articulated, street smoking bans were introduced mainly for environmental purposes such as littering prevention and connection with “beautification” [18]. At the national level, smoking is not restricted or prohibited by law in indoor public places, workplaces, or on public transportations. On the other hand, at the subnational level, two prefectures in Japan have implemented smoke-free ordinances for indoor public places with associated penalties for noncompliance. Specifically, due to the governor’s political leadership and intensive communication between the government and various stakeholders, Kanagawa Prefecture (one of the most populous prefectures in Japan) is the first prefecture that passed an ordinance to restrict smoking in indoor public places in 2009 and this ordinance was enforced in 2010. Hyogo Prefecture followed as the second prefecture to adopt a similar ordinance in March 2012 and enforced a smoking ban in April 2013 [19]. Until now, Kanagawa and Hyogo prefectures have been the only two prefectures to implement legislative smoking bans with penalties for noncompliance in Japan. One previous study revealed that the Hyogo smoking ban was associated with better health outcomes only in the capital city of Hyogo Prefecture (Kobe city) [20].

Responding to international calls for smoke-free games, the Japanese government approved its first national smoking ban inside public facilities on 18 July 2018. This ordinance will be implemented in phases with complete enforcement by April 2020 [21]. The new national law prohibits indoor smoking at schools, hospitals, and government offices. However, smoking will not be comprehensively prohibited in other public facilities, including restaurants and bars. For example, larger and new eateries (capitalized at more than 50 million yen and with floor space of larger than 100 m^2^) are allowed to set up segregated, well-ventilated rooms for smoking. Smaller eateries (capitalized at 50 million yen or less and with floor space of up to 100 m^2^) are exempted from the ban, and such small eateries represent more than half of eateries in Japan. This policy design is similar to the Hyogo smoking ban, and thus our study may be useful to predict the potential impact of the national smoking ban for the Tokyo 2020 Olympics.

To complement the literature on the effectiveness of partial smoking bans, this paper investigates the impact of a partial smoking ban in Hyogo Prefecture on nonsmokers’ SHS exposure in both public and private places using data from Japan. Our findings may provide useful implications for future tobacco control policies in other countries, particularly for smoking bans for the Tokyo 2020 Olympics in Japan.

## 2. Materials and Methods

### 2.1. Data Sources

We used nationally representative, population-based repeated cross-sectional data from the Comprehensive Survey of Living Conditions (CSLC), which was conducted by the Japanese Ministry of Health, Labour and Welfare (MHLW) from 2001, 2004, 2007, 2010, 2013, 2016. The CSLC has collected information about household characteristics and health conditions every three years in June since 1986. We only used the most recent six waves of data because the survey did not collect information about smoking intensity before 2001. The questionnaire on household and health covers all respondents, including approximately 800,000 individuals from 300,000 households randomly selected in each survey year.

In addition, we used data from the National Health and Nutrition Survey (NHNS), which was also a nationally representative, population-based repeated cross-sectional dataset collected by the Japanese MHLW, from 2010, 2013, 2016. The NHNS has collected information about health and nutritional intake annually each November since 1947. We only used three waves of data from 2010 to 2016 because we only have access to data of every three years up to 2016 and we do not have information about passive smoking before 2010. The NHNS also includes a portion of the CSLC respondents, including approximately 10,000 individuals. Thus, the NHNS was able to be linked with the CSLC.

### 2.2. Outcome Measures

We use self-reported exposure to SHS from the NHNS as a measure of passive smoking. On the questionnaire, there were several places to report passive smoking (household, workplace, school, restaurant, game hall, and others). Respondents (who were 20 years of age or older) were asked how often they had been exposed to SHS in each place. The measurement that was assessed was the frequency of exposure to SHS, which included (1) every day; (2) several times per week; (3) once per week; (4) once per month; or (5) no exposure. For easier interpretation, we convert these categorical outcomes into continuous numbers as follow: (1) every day is converted to 30 days per month; (2) several times per week is converted to 15 days per month; (3) once per week is converted to four days per month; (4) once per month is converted to one day per month; (5) no exposure is converted to 0 days per month.

Additionally, there was a choice of “do not go there” for all the locations except for the household, and we excluded respondents who reported that they did not go to that particular place in the estimation since they were unlikely to be exposed to SHS and they were not affected by the smoking ban if they did not go to the specific locations. Nevertheless, some may argue that exposure to SHS in some places such as restaurants or game halls could affect an individual’s probability of going to that location; as such, the choice “do not go there” might be related to the treatment variable, the Hyogo smoking ban. For example, people might go to restaurants more often if the smoking ban reduced SHS there. Moreover, although SHS hardly affects people’s propensity to work, high exposure to SHS in the workplace might lead workers to change their jobs. We assumed that the smoking ban did not influence whether people chose to go to the specific locations in our sample, and our robustness checks confirmed that the probabilities of whether respondents would go to a particular location were not associated with the implementation of the smoking ban (see Appendix A, Table A1).

As additional supporting evidence to our main results, we also investigated whether the Hyogo smoking ban affected people’s smoking behaviors in both extensive margins (smoking status) and intensive margins (smoking intensity). If the smoking regulation displaced smokers from indoor public places to indoor private places, individuals’ overall smoking behaviors should not be affected significantly. For smoking behaviors, we used data from the CSLC and examined two outcome variables: smoking status and smoking intensity. In the CSLC, for smoking status, respondents were classified into four categories; (1) nonsmoker (I do not smoke); (2) daily smoker (I smoke every day); (3) occasional smoker (I smoke occasionally but not every day); (4) quitter (I have stopped smoking for more than one month). Using these categories, we defined two indicators. First, a smoking indicator that measures smoking prevalence, it takes the value of 1 for categories (2) and (3), and the value of 0 for categories (1) and (4). Second, a quit indicator that measures smoking cessation, it takes the value of 1 for category (4), and the value of 0 for categories (2) and (3), in this case, nonsmokers were excluded. If the respondents were classified as categories (2) or (3), they were further asked how many cigarettes they smoke on average per day. For smoking intensity, using smokers’ daily cigarette consumption, smokers were classified into categories of 1–10, 11–20, 21–30, and ≥31 cigarettes. (descriptive statistics of smoking behaviors see Appendix A, Table A2).

### 2.3. Estimation Strategy

The study was a population-based, before-and-after observational study conducted under the framework of the difference-in-differences (DID) method. The treatment group consisted of respondents who lived in Hyogo Prefecture where a smoking ban was implemented in 2013, and the control group consisted of nonsmokers who lived in other prefectures where no such smoking bans were introduced. We excluded Kanagawa Prefecture from our analysis because data on the SHS exposure are not available in the NHNS before the implementation of the ban in Kanagawa prefecture in 2010. Due to this data limitation, we could not analyze the influence of the partial smoking ban on nonsmokers’ SHS exposure in Kanagawa Prefecture. In this study, the changes in SHS exposure among nonsmokers in Hyogo Prefecture were compared to the changes in SHS exposure among nonsmokers in other 45 prefectures without any smoking ban. To illustrate our difference-in-difference (DID) approach, we start from describing the following model.
(1)$$Y_{it}\;=\;\alpha \;+\;\beta_{1}Treat_{it}\;+\;\beta_{2}Post_{it}\;+\;\beta_{DID}\;\left(Treat_{it}\;\times \;Post_{it}\right)\;+\;X_{it}^{\prime }\beta_{X}\;+\;\varepsilon_{it}$$
where $Y_{it}$ measures the frequency of SHS exposure (measured in days have exposure to SHS per month) for respondent i at survey year t, $Treat_{it}$ is a dummy variable indicates the treatment (smoking ban), it equals 1 if respondent i is living in Hyogo Prefecture at year t and equals 0 otherwise. $Post_{it}$ is a dummy variable that indicates the post-treatment period, it equals 1 if the time t is 2013 or 2016 and equals 0 otherwise. $Treat_{it}\;\times \;Post_{it}$ is the interaction term between $Treat_{it}$ and $Post_{it}$, and $\beta_{DID}$ is our interest and measures the influence of the smoking ban on $Y_{it}$. $X_{it}^{’}\;$ is a vector of control variables including age, gender, household size, employment status, and occupation type. Additionally, in our estimation, we add a set of 44 prefecture dummies (Hokkaido prefecture is excluded as a reference group), a linear time trend, and their interaction terms step by step to capture unobserved prefecture fixed effects (e.g., geographic or weather characteristics), a linear time trend (e.g., nation-level anti-smoking trend over time like in Figure 1), and a prefecture-specific linear time trend (e.g., prefecture-specific anti-smoking trend over time), respectively. $\varepsilon_{it}$ is the error term that has a zero conditional mean and constant variance. $\varepsilon_{it}$ is the error term that has a zero conditional mean and constant variance.

To obtain a consistent estimator for $\beta_{DID}$, our DID approach needs to satisfy the following two assumptions. First, the common trend assumption (parallel trend assumption) requires that the outcomes show parallel trends between the control group and treatment group. Although we cannot test the validity of this identifying assumption directly by figure with only three time periods, we have three supporting facts for this assumption. First, we confirmed a parallel trend of smoking behaviors in both treatment and control groups before and after the intervention (year 2013) in Figure 1, this provides indirect support for the common trend assumption of SHS exposure since people’s smoking behavior did not differ significantly between treatment and control group before and after the smoking ban. Second, respondents in the treatment group and the control group were faced with the same tobacco price and consumption tax. Thus, we may reasonably expect that their smoking behaviors were not substantially different. Third, the NHNS conducted survey in November annually, and the survey time does not vary across different regions. Thus, respondents’ preferences would not be influenced by survey time.

Second, the DID approach assumes that there were no other policy changes or regional shocks that would affect individuals’ exposure to SHS when the Hyogo smoking ban was introduced. Although cigarettes price and other anti-smoking policies like a tobacco tax hike could also influence smoking behaviors and exposure to SHS, these policy changes were applied to the entire country and cigarette prices are uniform across all over Japan. Thus, we may reasonably expect that there were no such changes that influenced only Hyogo Prefecture.

For estimation, in our DID setup, we used ordinary least square (OLS) regression to estimate the parameters of interest. We controlled for individual socioeconomic characteristics, including age, sex, household size, employment status, and occupation type. All estimations were conducted using Stata 15.1(StataCorp LLC, Texas, USA).

## 3. Results

### 3.1. Descriptive Statistics

Table 1 reports the descriptive statistics of the analytical sample. Overall, the frequency of having passive smoking shows a declining trend, nonsmokers in the treatment group on average have 3.04 days per month of exposure to SHS in household and nonsmokers in the control group on average have 3.05 days per month of exposure to SHS in household. Among all the locations, nonsmokers have the most exposure to SHS in workplaces, 4.45 days per month for the treatment group and 5.18 days per month for the control group.

### 3.2. Secondhand Smoke Exposure

Table 2 summarizes our estimation results. The first, second, and third column present the effects on the SHS exposure in households, workplaces, and restaurants, respectively. In our estimation, we start from a simplest model with only main effects and add prefecture dummies and a linear time trend variable step by step. The first panel in Table 2 presents the results for the model with only main effects (Model 1). The second panel presents the results for the model with prefecture dummies (Model 2). The third panel presents the results for the model with prefecture dummies, a linear time trend, and prefecture-specific linear time trend (i.e., the interactions between prefecture dummies and a linear time trend) (Model 3). The first raw in each panel (i.e., Treat × Post ($\beta_{DID}$)) measures the effect of the smoking ban on the SHS exposure in each location.

While the sign of the coefficients is consistent across all the models, the magnitude of $\beta_{DID}$ tends to be larger in Model 3 than in Models 1 and 2. For example, the sign of $\beta_{DID}$ is positive in all locations for all models, which the magnitude of $\beta_{DID}$ in Model 3 is more than double of that in Models 1 and 2 for all locations. These results may imply that controlling for a linear time trend such as increasing anti-smoking is important to capture the effect of the smoking ban on the SHS exposure. Thus, we hereafter focus on the results in Model 3.

Our results in Model 3 show that the partial smoking ban in Hyogo Prefecture increased nonsmokers’ exposure to SHS in households and workplaces by 2.64 days (95% CI 0.108–5.167) and 4.70 days (95% CI 0.978–8.417) per month, respectively, and the impacts of the partial smoking ban on nonsmokers’ exposure to SHS in households and workplaces are statistically significant at the 5% level. In contrast, the smoking ban had no significant effect on nonsmokers’ exposure to SHS at restaurants.

### 3.3. Smoking Behaviors

As shown in Table 3, neither smoking status nor smoking intensity were significantly affected by the partial smoking ban (Figure 1 also provides some visual evidences). The results imply that respondents in Hyogo Prefecture did not change their smoking behaviors under the restriction of the smoking ban. These results provide indirect support for our main results that exposure to SHS did not change in indoor public places (restaurants) while it increased in indoor private places (workplaces and households).

## 4. Discussion

This study examined the impact of a partial smoking ban on nonsmokers’ exposure to SHS in Japan. Hyogo Prefecture implemented a legislative smoking ban with a corresponding penal code in 2013, whereas all other prefectures in Japan, except for Kanagawa Prefecture, have never implemented such smoking bans until now. We exploited this regional policy change as a natural experiment to identify the association between the partial smoking ban and nonsmokers’ exposure to SHS in both public and private places. We employed a DID framework, using nationwide data from the NHNS for the years 2010, 2013, and 2016. We found a significant increase in nonsmokers’ exposure to SHS in households and workplaces after the implementation of the smoking ban, while nonsmokers’ exposure to SHS at restaurants did not change significantly. This may be because numerous small eateries were exempted from the smoking ban, and smokers could go to these non-regulated restaurants and smoke there, thus the exposure to SHS did not change at restaurant on average.

One main concern of our model specification is over-specification due to the control of a full set of prefecture dummies and prefecture-specific linear time trend given that our empirical analysis for SHS exposure only includes three time periods (2010, 2013, and 2016). We have three reasons for justification. First, we could not include year dummies (i.e., take the value of 1 for a specific year) because they over-control the effect of smoking ban (multicollinearity problem); thus, as a second-best way to control for some unobserved time changes, we include a linear time trend variable (i.e., changes from 1 to 3 over the three years). Since the effect of the smoking ban is discrete and nonlinear, we believe the linear time trend does not over-control the effect of smoking ban. Second, smoking behaviors vary across prefectures in Japan, for example, in our CSLC (2001–2016) sample, the average smoking prevalence was 25.13%, while Hokkaido Prefecture had the highest smoking prevalence of 31.99% and Nara Prefecture had the lowest smoking prevalence of 22.02%, omitting prefecture fixed effects and prefecture-specific time trend would lead to biased results. Third, the estimation results in Table 2 show that controlling for prefecture fixed effects and prefecture-specific linear time trend affected our results substantially, these results imply that socioeconomic characteristics (e.g., culture, diet) that may affect smoking patterns are different across prefectures, it is necessary to include prefecture dummies to capture the regional fixed effects given that we could not control individual income or expenditure (the NHNS data do not contain income or expenditure information).

If other prefecture-level tobacco control policies that might influence smokers’ smoking behaviors and nonsmokers’ exposure to SHS were implemented concurrently with the Hyogo smoking ban, our main estimation results would be confounded. However, no such policy changes occurred during the period of 2010–2016. Japan has had a Tobacco Business Act since 1984, and this law restricts the way of producing, retailing, and retail prices. Under the law, the retailers have to sell tobacco at the list price and are not allowed to change the tobacco price, such as providing discount. Hence, cigarettes prices do not vary across prefectures or regions in Japan, and the price of a particular brand of cigarettes is the same across all vendors, from cigarette machines to big supermarkets. Moreover, there are no discounts for bulk purchases. All taxes on cigarettes, such as consumption sales tax and tobacco tax, are uniform across prefectures. The legal age for smoking is 20 years old in Japan, and it did not change during our study period. Although Japan introduced a tobacco tax increase in October 2010, this tax hike was uniform throughout the country. Thus, its effect should be captured by our time dummy variables.

Similar to the results obtained by Adda and Cornaglia (2010) and Ho et al., (2010) [16,17], our findings support the displacement hypothesis, which remains controversial in literature on smoking bans. Although a meta-analysis indicated that most studies have confirmed that a public smoking ban reduced exposure to SHS in childrens’ homes [22], our results may still be suggestive because the partial smoking ban in Japan exempted workplace and numerous restaurants. Related studies about Japan suggest that complete smoking bans in workplace were associated with improved smoking behaviors and health [23] while partial and no bans were associated with high SHS exposure [24]. As stated by Yamada et al. (2015) [19], the Hyogo smoking ban failed to provide effective protection against SHS exposure because the ordinance mentioned only SHS in public places and ignored SHS in workplaces. This was because workplaces are covered by the Industrial Safety and Health Law (ISHL) rather than the department of health. This also explains why we found that nonsmokers’ exposure to SHS increased in workplaces after the smoking ban.

It is also worth noting that there are four limitations that might influence our conclusions. First, we used the frequency of exposure to SHS as the best available measure for SHS exposure, but it is not as precise as biomarker data such as cotinine concentration. Considering that self-reported SHS exposure measures only recognized exposure while biomarker data may also measure unrecognized SHS exposure, our results may be considered as the lower limit of the effect of the partial smoking ban on nonsmokers’ SHS exposure. Second, children are typically the most vulnerable group of people who might be exposed to SHS; however, the questions about SHS exposure in the NHNS were only asked to adults 20 years of age or older at that time. Future research can explore how a partial smoking ban affects SHS exposure of children in Japan. Third, regarding smoking cessation, we only have information about quitters who have stopped smoking for more than one month, we could not observe the full past use of smoking and the possibility of relapse. Fourth, due to data limitations, the pretreatment group included only 273 nonsmokers and we have only one period pre-treatment, which makes our findings less representative of the total population. Thus, we treat our findings as only suggestive evidence rather than conclusive evidence.

Despite the limitations, our findings may still provide two important policy implications. First, the government should promote risk awareness of nonsmokers’ SHS exposure in indoor private places such as households when smoking is prohibited in indoor public places. For example, the Hyogo smoking ban was found to increase nonsmokers’ SHS exposure in workplaces and households, which was arguably because workplaces and households were exempted from the ban. Similarly, although the new national smoking ban for Tokyo 2020 Olympics prohibits smoking at workplaces in addition to restaurants, it may increase nonsmokers’ exposure to SHS in households. Since comprehensive smoking bans have a stronger effect on policy support than partial smoking bans [25,26], a more comprehensive smoking ban is needed to curb smoking and protect nonsmokers in Japan. Second, if a comprehensive smoking ban is not feasible, an increase in nonsmokers’ SHS exposure may be mitigated by combining a partial smoking ban with tobacco tax increases. This is because excise taxes have been found to be effective in curbing passive smoking. For example, previous studies have found that tobacco tax increases effectively reduced nonsmokers’ SHS exposure [16] and smokers’ cessation rate [27]. Thus, combining partial smoking bans and tobacco tax increases might be the second-best strategy to curb nonsmokers’ SHS exposure.

## 5. Conclusions

Based on the nationally representative data from Japan, this paper investigated the impact of a partial smoking ban introduced in Hyogo Prefecture in 2013 on nonsmokers’ exposure to secondhand smoke (SHS) by employing the difference-in-differences approach. We found that a partial smoking ban may hardly change overall smoking behaviors and rather displace smokers from regulated public places to nonregulated private places, which can increase nonsmokers’ exposure to SHS at private places (e.g., workplaces, homes). At least in Japan, future tobacco control policies may need to consider complete smoking bans instead of partial smoking bans to better protect nonsmokers from SHS exposure.

## Figures and Tables

**Figure 1 ijerph-16-02804-f001:**
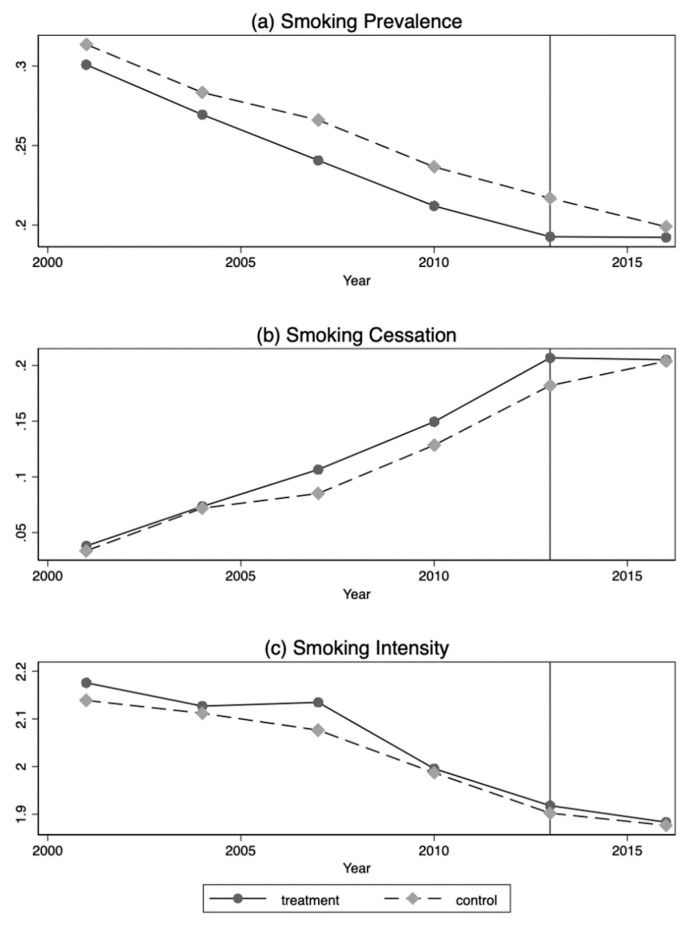
Smoking behaviors (Source: CSLC 2001–2016). Smoking prevalence (respondents who were daily smoker or occasional smoker, a dummy variable, yes = 1, no = 0), smoking cessation (smokers who have stopped smoking for more than one month, a dummy variable, yes = 1, no = 0), and smoking intensity (number of cigarettes smoked per day for smokers, a categorical variable, 1–10 cigarettes per day = 1, 11–20 cigarettes per day = 2, 21–30 cigarettes per day = 3, ≥31 cigarettes per day = 4).

**Table 1 ijerph-16-02804-t001:** Descriptive statistics of analytical sample for passive smoking.

Passive Smoking (%)	Full			Pre-Treatment			Post-Treatment	
	Treat	Control		Treat	Control		Treat	Control
	n = 1003	n = 29241		n = 273	n = 5064		n = 730	n = 24177
Household								
Every day	8.37	8.43		10.62	9.99		7.53	8.10
Several times per week	3.19	2.82		1.83	3.38		3.70	2.71
Once per week	0.80	1.88		0.73	2.29		0.82	1.80
Once per month	1.99	2.21		3.30	2.88		1.51	2.07
No exposure	85.64	84.65		83.52	81.46		86.44	85.32
Frequency (day per month)	3.04	3.05		3.52	3.62		2.86	2.93
	(4.45)	(8.56)		(9.37)	(9.21)		(8.26)	(8.41)
Workplace								
Every day	4.89	6.56		4.76	8.85		4.93	6.08
Several times per week	5.88	6.24		8.42	7.01		4.93	6.08
Once per week	3.49	2.96		4.76	3.67		3.01	2.81
Once per month	3.79	3.52		5.86	3.79		3.01	3.46
No exposure	38.68	39.79		36.63	35.25		39.45	40.74
Do not go there	43.27	40.93		39.56	41.43		44.66	40.82
Frequency (day per month)	4.45	5.18		4.87	6.64		4.28	4.87
	(9.07)	(9.89)		(8.97)	(10.95)		(9.11)	(9.63)
School								
Every day	0.00	0.13		0.00	0.14		0.00	0.13
Several times per week	0.40	0.26		0.73	0.22		0.27	0.27
Once per week	0.30	0.19		0.00	0.22		0.41	0.19
Once per month	0.60	0.32		0.37	0.36		0.68	0.32
No exposure	20.14	19.62		16.85	16.96		21.37	20.18
Do not go there	78.56	79.49		82.05	82.11		77.26	78.92
Frequency (day per month)	0.36	0.44		0.63	0.48		0.28	0.43
	(2.08)	(2.94)		(3.00)	(3.11)		(1.72)	(2.91)
Restaurant								
Every day	0.70	0.53		0.37	0.63		0.82	0.51
Several times per week	2.69	2.66		2.56	3.32		2.74	2.52
Once per week	8.67	6.25		12.82	6.87		7.12	6.12
Once per month	23.13	19.27		27.84	20.06		21.37	19.10
No exposure	35.79	39.50		30.77	33.02		37.67	40.85
Do not go there	29.01	31.79		25.64	36.10		30.27	30.89
Frequency (day per month)	1.68	1.47		1.73	1.82		1.66	1.40
	(4.11)	(3.93)		(3.52)	(4.39)		(4.32)	(3.82)
Game hall								
Every day	0.30	0.24		0.00	0.22		0.41	0.24
Several times per week	1.20	1.51		0.73	1.88		1.37	1.44
Once per week	1.99	2.51		2.20	2.59		1.92	2.49
Once per month	2.69	3.84		2.20	4.36		2.88	3.73
No exposure	16.15	15.62		15.38	15.56		16.44	15.63
Do not go there	77.67	76.28		79.49	75.39		76.99	76.46
Frequency (day per month)	1.68	1.85		1.07	2.01		1.89	1.81
	(4.79)	(4.68)		(2.98)	(4.79)		(5.25)	(4.65)
Controlled covariates								

Age	56.51	58.14		52.82	57.25		57.88	58.33
	(18.13)	(17.99)		(16.08)	(17.65)		(18.66)	(18.05)
Household size	2.88	2.91		3.09	3.03		2.80	2.88
	(1.33)	(1.40)		(1.35)	(1.43)		(1.32)	(1.39)
Gender (Male = 1)	0.39	0.39		0.40	0.39		0.39	0.39
	(0.49)	(0.49)		(0.49)	(0.49)		(0.49)	(0.49)
Employment status (Employed = 1)	0.63	0.68		0.65	0.69		0.62	0.67
	(0.48)	(0.47)		(0.48)	(0.46)		(0.49)	(0.49)
Occupation type (%)								
Technological	10.47	10.92		13.19	10.21		9.45	11.07
Management	3.99	2.99		5.49	3.02		3.42	2.99
Officer	11.47	9.28		12.45	9.14		11.10	9.31
Salesperson	3.79	4.85		4.40	5.02		3.56	4.82
Service	5.28	7.96		5.13	8.18		5.34	7.92
Security guard	0.50	0.70		1.47	0.65		0.14	0.72
Agriculture	2.79	3.70		2.56	4.40		2.88	3.55
Machine operation	1.50	1.10		2.20	1.46		1.23	1.03
Production process	6.38	7.75		5.86	8.04		6.58	7.68
housework	28.22	22.72		28.57	25.20		28.08	22.21
Others	16.75	18.29		12.09	18.40		18.49	18.27
Students	8.87	9.73		6.59	6.28		9.73	10.45

Notes: Standard deviation are reported in parentheses. Data source: National Health and Nutrition Survey (NHNS 2010, 2013, 2016).

**Table 2 ijerph-16-02804-t002:** The impact of a partial smoking ban on nonsmokers’ frequency of exposure to secondhand smoke (SHS) (measured by days of having exposure to SHS per month).

	Household	Workplace	Restaurant
Model 1	(1)	(2)	(3)
Treat × Post ($\beta_{DID}$) Standard Error (SE)	0.156	1.472 *	0.425
	(0.642)	(0.796)	(0.316)
95% CI	[−1.102, 1.413]	[−0.087, 3.032]	[−0.193, 1.044]
Treat ($\beta_{1}$)	−0.132	−1.946 ***	−0.169
SE	(0.568)	(0.660)	(0.252)
95% CI	[−1.246, 0.982]	[−3.240, −0.653]	[−0.663, 0.326]
Post ($\beta_{2}$)	−0.580 ***	−1.655 ***	−0.412 ***
SE	(0.137)	(0.206)	(0.082)
95% CI	[−0.848, −0.312]	[−2.060, −1.251]	[−0.572, −0.252]
Adjusted R^2^	0.045	0.093	0.031
F-statistics	83.23	123.57	30.52
N	30244	17843	20657
Model 2 (with Prefecture dummies)	Household	Workplace	Restaurant
Treat × Post ($\beta_{DID}$)	0.159	1.517 *	0.253
SE	(0.643)	(0.798)	(0.316)
95% CI	[−1.101, 1.419]	[−0.048, 3.081]	[−0.367, 0.874]
Treat ($\beta_{1}$)	−0.485	−3.090 ***	0.073
SE	(0.641)	(0.802)	(0.303)
95% CI	[−1.741, 0.772]	[−4.663, −1.518]	[−0.520, 0.667]
Post ($\beta_{2}$)	−0.586 ***	−1.705 ***	−0.239 ***
SE	(0.141)	(0.212)	(0.084)
95% CI	[−0.862, −0.311]	[−2.120, −1.289]	[−0.404, −0.073]
Adjusted R^2^	0.047	0.096	0.042
F-statistics	25.29	36.93	12.60
N	30244	17843	20657
Model 3 (with Prefecture dummies and a linear time trend)	Household	Workplace	Restaurant
Treat × Post ($\beta_{DID}$)	2.638 **	4.697 **	1.049
SE	(1.292)	(1.898)	(0.904)
95% CI	[0.108, 5.167]	[0.978, 8.417]	[−0.723, 2.821]
Treat ($\beta_{1}$)	1.101	− 2.340	0.212
SE	(1.174)	(1.735)	(0.768)
95% CI	[−1.199, 3.402]	[−5.742, 1.062]	[−1.293, 1.718]
Post ($\beta_{2}$)	0.352	−0.833 **	0.322 *
SE	(0.264)	(0.385)	(0.166)
95% CI	[−0.165, 0.870]	[−1.588, −0.078]	[−0.002, 0.647]
Adjusted R^2^	0.049	0.098	0.044
F-statistics	15.32	21.92	7.91
N	30244	17843	20657

Notes: Controlled covariates include age, age square, gender, household size, employment status, and occupation type. Heteroskedasticity-robust standard errors are reported in parentheses. 95% confidence intervals are presented in brackets. * Inference: * *p* < 0.1; ** *p* < 0.05; *** *p* < 0.01. Data source: National Health and Nutrition Survey (NHNS 2010, 2013, 2016).

**Table 3 ijerph-16-02804-t003:** The impact of a partial smoking ban on smoking status and smoking intensity.

	Smoking Status		Smoking Intensity (Cigarettes/Day)
	Smoking Prevalence	Smoking Cessation	
	(1)	(2)	(3)
Treat × Post ($\beta_{DID}$)	0.010	−0.014	0.002
SE	(0.006)	(0.011)	(0.027)
95% CI	[−0.002, 0.022]	[−0.036, 0.007]	[−0.050, 0.054]
Treat ($\beta_{1}$)	−3.239 **	−4.235 **	1.549
SE	(1.348)	(1.839)	(5.117)
95% CI	[−5.881, −0.597]	[−7.840, −0.630]	[−8.481, 11.578]
Post ($\beta_{2}$)	0.006 ***	0.026 ***	−0.057 ***
SE	(0.001)	(0.002)	(0.004)
95% CI	[0.004, 0.008]	[0.023, 0.029]	[−0.064, −0.049]
Adjusted R^2^	0.179	0.063	0.118
F-statistics	4836.45	333.28	718.12
n	2,366,896	672,879	592,551

Notes: Column (1) and column (2) correspond to results estimated from linear probability models, column (3) corresponds to results estimated from ordinary least square model. Controlled covariates include age, age square, gender, household size, household expenditure, marital status, self-rated health, employment status, occupation type, a linear time trend, prefecture fixed effects, and linear form prefecture-specific time trend. Heteroskedasticity-robust standard errors are reported in parentheses. 95% confidence intervals are presented in brackets. * Inference: * *p* < 0.1; ** *p* < 0.05; *** *p* < 0.01. Data source: Comprehensive Survey of Living Conditions (CSLC 2001, 2004, 2007, 2010, 2013, 2016).

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
