# Peer review of "Partial Smoking Ban and Secondhand Smoke Exposure in Japan"

_ijerph, 2019, doi:10.3390/ijerph16152804_

Round 1

Reviewer 1 Report

PREMISE
Largely a well-written article.  However, I am not convinced that the findings make sense. The authors need to describe and justify their analytic choices.

ABSTRACT
Abstract should be more specific /informative and writing/phrasing should be more succinct (don't need words like 'to examine this possibility'). Methods need to indicate study years, datasets, minimum sample size, main outcome variable(s).

INTRODUCTION
Recommend stronger focus on background for Japan to understand the context there.
--What exactly does 'partial' ban mean in this context?  Authors say 'indoor public places' but do not specifically say what this means.  Are restaurants included because the public can patronize a restaurant? Or only train stations that are obviously public? Is the 'partial' ban the same as what the new national law is?
'...restrict smoking in indoor public places in 2009 and implemented a legislative smoking ban in 2010' what does this mean? What is the difference (restrict.. legislative ban')
--How large is Kanagawa.  Why did they have the political will to implement a ban (it is progressive / health-focused city etc?).
--If the sample went up through 2015, then why wasn't Hyogo in the treatment group (bans implemented in 2012)?

MIS-SPECIFIED or LACK OF SPECIFICITY
--Interaction mis-specified.
Tables on line 120, 192, 205, 202-- the authors want to exam interaction (for example treatment x time).  But in order to interpret the interaction, they need to include in the model -- and show in the table -- the main effects (in this example the main effects for treatment and time). If the authors alter from this standard, then they need to provide justification. (RCT can sometimes do away with the main effects if they are assumed zero, but observational studies cannot.)  Please consult an epidemiology / statistics book.

--Model may be over-specified.  The data are only a few years and yet the authos include a "full set of prefecture dummies to capture the time fixed effects and prefecture fixed effects". The authors need to explain why this is not over-specified.

--Exposure or Outcome not specified.
Table on line 205 - title needs to indicate how exposure to patial smoking ban is operationlized. Does exposure mean 'treatment' group and thus exposed is only Kanagawa prefecture?
Table on line 220-- how the outcome is operationalized.  The descriptive table shows the outcome responses (every day, several times per week, etc) but the regression table 3 does not indicate what the outcome is.

--Only a single year post-ban is not much to compare the pre-ban period with. This should be mentioned in the limitations. Has the survey been collected 2016-2018? 

--Line 199.  Were restaurants covered?  If yes, then it seems implausible that restaurant workers did not experience reduction in SHS.  Unless their worksites had already banned smoking

SMALL COMMENTS
Quitting.  Past use of smoking is not well-specified. The Methods suggest that there is only information on quitting in past month ('quitter (I have stopped smoking for more than one month)'). The authors should explain why they do not think this is a major limitation.

Pricing. My opinion is that it is very unusual that prices are uniform across a country. Please provide a reference or a couple of words that explain the situation. Is this due to gov't mandate or gov' subsidies.

Line 67. This sentence doesn't make sense.  The strength of this article is that there is a control group (no ban), not national representativenss.

Lines 68--85 could be moved to results and discussion. In social sciences, the results are often put into the introduction but that is not the norm in public health. I leave this up to the editor to decide.

Appendix tables are referred to but it looks like the authors are referring to tables in the main manuscript and then two tables are called Table 1 etc.

Table titles need to provide more detail so that they 'stand alone'. 

Line 98. check punctuation.

Author Response

Dear Reviewers

Thank you for your valuable comments. We have carefully reviewed your comments and have revised the manuscript accordingly. We replied to the comments in a point-by-point manner below. Our responses are presented in red. We also attached the manuscript with the records of changes for your reference. Please note that the page and line numbers in our replies are those in the revised manuscript without displaying the records of changes.

Replies to Reviewer 1:

ABSTRACT
Abstract should be more specific /informative and writing/phrasing should be more succinct (don't need words like 'to examine this possibility'). Methods need to indicate study years, datasets, minimum sample size, main outcome variable(s).

We revised the abstract following all the comments, specifically, we deleted redundant words, added more information about the study years, datasets, sample size, and outcome measures. (Page 1 line 13 to line 24)

INTRODUCTION
Recommend stronger focus on background for Japan to understand the context there. 
--What exactly does 'partial' ban mean in this context?  Authors say 'indoor public places' but do not specifically say what this means.  Are restaurants included because the public can patronize a restaurant? Or only train stations that are obviously public? Is the 'partial' ban the same as what the new national law is? 

We added more explanations following the questions. A partial smoking ban means a smoking ban with exemptions or designated smoking rooms (Page 1 line 34). In the context of Hyogo prefecture in Japan, the partial smoking ban prohibits smoking in indoor public places like schools, hospitals, government facilities, and public transport facilities etc. However, private businesses such as restaurants, bars, hotels, and leisure facilities were able to choose either to prohibit smoking or introduce smoking separation. The new national law has a very similar policy design, it prohibits indoor smoking at schools, hospitals and government offices, while larger and new eateries (capitalized at more than 50 million yen and with floor space of larger than 100 square meters) are allowed to set up segregated, well-ventilated rooms for smoking. Smaller eateries (capitalized at 50 million yen or less and with floor space of up to 100 square meters) are exempted from the ban, and the smaller eateries represent more than half of eateries in Japan (Page 2 line76).

'...restrict smoking in indoor public places in 2009 and implemented a legislative smoking ban in 2010' what does this mean? What is the difference (restrict.. legislative ban')

We realized that our previous expression was misleading, so we changed the sentence as follow: “Kanagawa Prefecture is the first prefecture that passed an ordinance to restrict smoking in indoor public places in 2009 and this ordinance was enforced in 2010.” (Page 2 line 65)

--How large is Kanagawa.  Why did they have the political will to implement a ban (it is progressive / health-focused city etc?). 

Kanagawa is one of the most populous prefectures in Japan, the rapid enactment of the smoking ban was benefited from the political leadership of the governor as well as intensive communication between the government and a wide range of stakeholders (Page 2 line 62).

--If the sample went up through 2015, then why wasn't Hyogo in the treatment group (bans implemented in 2012)?

Hyogo Prefecture is the treatment group, the ban was implemented in April 2013 while our data from the National Health and Nutrition Survey (NHNS) 2013 was conduct in November, so we treated year 2013 as the post-treatment period (page 4 line 145).

MIS-SPECIFIED or LACK OF SPECIFICITY 
--Interaction mis-specified. 
Tables on line 120, 192, 205, 202-- the authors want to exam interaction (for example treatment x time).  But in order to interpret the interaction, they need to include in the model -- and show in the table -- the main effects (in this example the main effects for treatment and time). If the authors alter from this standard, then they need to provide justification. (RCT can sometimes do away with the main effects if they are assumed zero, but observational studies cannot.)  Please consult an epidemiology / statistics book.

        Thank you for pointing this out, we included and reported the main effects coefficients in the revised version (please see each table).

--Model may be over-specified.  The data are only a few years and yet the authors include a "full set of prefecture dummies to capture the time fixed effects and prefecture fixed effects". The authors need to explain why this is not over-specified.

        Page 9 line 257: We conducted additional robustness checks (the third reason below), and we believe that prefecture dummies are important and that our results are robust to over-specification, if any, because of the following three reasons. First, we have 30,244 observations and 46 prefectures (Kanagawa Prefecture was excluded) in total, hence, we still have enough degree of freedom after including prefecture dummies and prefecture-specific time trend. Second, the over-specified model leads to inflated standard errors while estimates remain unbiased. Moreover, past studies [10, 16] with similar designs also controlled state dummies and state-specific time trend. Third, we also estimated two different specifications as robustness checks: (1) a model without prefecture dummies and prefecture-specific time trend, the results are reported in the Appendix A table A3; (2) a model with region dummies (12 regions in Japan) and region-specific time trend (instead of 47prefecture dummies), the results are presented in Appendix A table A4. The estimation results indicate that the partial smoking ban had no significant effects on nonsmokers’ exposure to SHS at any locations (households, workplaces, restaurants), the adjusted R square also decreased. These results imply that socioeconomic characteristics (e.g., culture, diet) that may affect smoking patterns are different across prefectures, it is necessary to include prefecture dummies to capture the prefecture fixed effects given that we could not control individual income or expenditure (the NHNS data do not contain income or expenditure information).

--Exposure or Outcome not specified. 
Table on line 205 - title needs to indicate how exposure to partial smoking ban is operationalized. Does exposure mean 'treatment' group and thus exposed is only Kanagawa prefecture? 

We realized that the original title was misleading, the exposure means exposure to secondhand smoke, we changed it to ‘passive smoking’ to make it clearer (page 6 line 204). The treatment group is Hyogo Prefecture, and other prefectures are controlled group. Although Kanagawa Prefecture was the first sub-nation to introduce a legislative smoking ban in Japan, because data on the secondhand smoke exposure is not available in the NHNS before the implementation of the ban in Kanagawa Prefecture in 2010. Due to this data limitation, we could not analyze the influence on the SHS exposure in Kanagawa Prefecture. Thus, we excluded Kanagawa Prefecture in our analysis because the impact of the smoking ban in Kanagawa would contaminate our control group. (page 4 line 145)

Table on line 220-- how the outcome is operationalized.  The descriptive table shows the outcome responses (every day, several times per week, etc) but the regression table 3 does not indicate what the outcome is.

The outcome responses in the descriptive table are categorical (every day; several days per week; several days per month; once per month; no exposure), and we were supposed to use ordered logit/probit models or multinomial logit/probit models. However, under the framework of difference-in-differences, interaction terms in nonlinear models cannot be explained in the same way as in linear models, and it is very difficult to justify the parallel trend for the transformed log-odds. Thus, we converted the categorical variable into continuous numbers (frequency of having exposure to secondhand smoke in days per month). We added explanations about how to convert the categorical responses into continuous numbers (page 3 line 112). We also reported the measured outcome we used for regression in the descriptive table, see table 1 (page 6 line 204).

--Only a single year post-ban is not much to compare the pre-ban period with. This should be mentioned in the limitations. Has the survey been collected 2016-2018? 

We have two years post-ban period and one-year pre-ban period, and we acknowledged this limitation in the discussion part (page 10 line 306). For the survey period, despite the survey conducts annually, we only have access to data of every three years up to 2016 (page 3 line 100).

--Line 199.  Were restaurants covered?  If yes, then it seems implausible that restaurant workers did not experience a reduction in SHS.  Unless their worksites had already banned smoking

Because the ban only covered some restaurants, and even these covered restaurants could choose either to prohibit smoking or set up designated smoking areas. Given that smoking in prohibited restaurants might decrease while smoking in the non-prohibited restaurants might increase (smokers might concentrate in these restaurants where smoking was still allowed), the overall SHS might possibly remain unchanged (page 8 line 248).

SMALL COMMENTS
Quitting.  Past use of smoking is not well-specified. The Methods suggest that there is the only information on quitting in the past month ('quitter (I have stopped smoking for more than one month)'). The authors should explain why they do not think this is a major limitation.

We acknowledged this as another limitation in the discussion part (page 10 line 304).

Pricing. My opinion is that it is very unusual that prices are uniform across a country. Please provide a reference or a couple of words that explain the situation. Is this due to gov't mandate or gov' subsidies.

Japan has a Tobacco Business Act since 1984. This law restricts the way of producing, retailing, and retail prices. Under the law, the retailers have to sell tobacco at the list price and are not allowed to change the price such as discounting (page 9 lines 274-279).

Line 67. This sentence doesn't make sense.  The strength of this article is that there is a control group (no ban), not national representativeness.

We deleted the unreasonable sentences.

Lines 68--85 could be moved to results and discussion. In social sciences, the results are often put into the introduction but that is not the norm in public health. I leave this up to the editor to decide.

Following the suggestion, we moved these paragraphs to the results and discussion sections.

Appendix tables are referred to but it looks like the authors are referring to tables in the main manuscript and then two tables are called Table 1 etc.

This was a mistake. We now corrected it and moved Table A1 and Table A2 to the Appendix.

Table titles need to provide more detail so that they 'stand alone'. 

We added more details in the table titles (please see each table).

Line 98. check punctuation.

We deleted wrong punctuation.

Reviewer 2 Report

The topic of this manuscript is very important with regard to present partial SHS bans are not effective. However, the design of manuscript should be improved. Because, there are overlapping in each sections.

1-      The current introduction is like a brief report which includes some results and discussions. Page 2 Line 68: This paragraph should be presented in the result section. Page 2 Line 74: This paragraph should be moved to the discussion. Page 2 Line 59: This paragraph should be moved before the paragraph on the line 48.

2-      It is so confusing that what is the exact sample size for this study. Which variable is coming from which datasets? Each table has different sample numbers. It should be about using different datasets, however, it should clearly be explained which dataset used in each table.

3-      There are two ‘Table 1’ in the manuscript (on Page 3 and Page 6).

4-      It is so hard to understand the numbers in the Tables. Please make them ease to understand as done in the article below. I do not understand which one is mean or β. For example, the first row in the Table 2 on page 7, if the values (2.813, 4.655, 1.223) are β coefficients, it has to clearly be mentioned in the table. In connection with this feedback, it would be good to state the significance range of confidence interval for DID analysis. For example, in the abstract, the stated numbers for the DID analyses seem like ratio, however, the CIs for these numbers include “1” and causing a confusion at first glance.

https://www.ncbi.nlm.nih.gov/pmc/articles/PMC4937082/pdf/40258_2016_Article_249.pdf

5-      Page 7 line 198: The following statement should be in the method section: “the positive impacts were statistically significant at the 5 % level”

6-      Page 7 line 211: This paragraph should be moved to the method section.

7-      Page 8 Line 227: This paragraph should be moved to the discussion section.

8-      Page 6 Line 186: The first paragraph of the result section should be moved to the method. Because, in the method section, the authors mention the `treatment and control` group so much. However, in order to understand who is treatment and control group, I had to keep reading until result section.

9-      In the result section, there are so much interpretations which had to be done in the discussion. I would prefer to read results only in the result section and interpretation in the discussion.

10-   In the refs numbered 16, 17, and 22, the colon was used to separate issue and page numbers which inconsistent with other refs.

Round 2

Reviewer 1 Report

Thank you for your revised manuscript.

More clarification is needed. In some cases the authors clarified in the response to the reviewers but barely clarified in the text of the manuscript. See suggested clarifications in caps

line 82.  this paper investigates the impact of a partial smoking ban in HYOGO PREFECTURE on nonsmokers....

line 139. Need to make the fact that they subset the population to nonsmokers. why are the authors saying 'we mainly focused on smoking prevalence and smoking cessation."  They were also interested in NONSMOKERS (which isn't the same as smoking prevalence).

line 147.  the trt group consisted of NONSMOKERS who lived in...

line 148   the control group consisted of NONSMOKERS who lived in other prefectures where no smoking ban was introduced.  WE EXCLUDED KANAGAWA PREFECTURE BECAUSE DATA ON THE SECONDHAND SMOKE EXPOSURE IS NOT AVAILABLE IN THE NHNS BEFORE THE IMPLEMENTATION OF THE BAN IN KANAGAWA PREFECTURE IN 2010. DUE TO THIS DATA LIMITATION, WE COULD NOT ANALYZE THE INFLUENCE ON THE SHS EXPOSURE IN KANAGAWA PREFECTURE. THUS, WE EXCLUDED KANAGAWA PREFECTURE IN OUR ANALYSIS BECAUSE THE IMPACT OF THE SMOKING BAN IN KANAGAWA WOULD CONTAMINATE OUR CONTROL GROUP.

Sentence line 254 should be deleted.  The 'positive coefficent' cannot be interpreted for restaurants. The authors can see that the confidence interval is very wide -- uncertainty is too great.  The direction of the estimate could be negative or positive.

line 366.  table 3 or is this Appendix table 3??  Confusion over main and appendix tables was brought up by both reviewers.

Overall, the SHS results still do not fully make sense.  The findings indicate that the bans actually increased SHS exposure at workplaces over and above the areas that did not have bans.  I believe the results in Appendix Table 3 (line 365s onward) are more intuitive. The appendix results indicate that overall SHS exposure was lower in Hyogo but SHS did not decline post-ban in Hyogo.
Regarding the issue of overspecification. Can you provide evidence that the model with just main effects (not interaction) and all the dummies and covariates does not suffer from multi-collinearity?  Can the authors provide multicollinarity diagnostics for that model?  
I am not convinced by the rationale that the authors put forward lines 260..274.  They don't mention that over-specification leads to uniterpretable results. I agree that it would be good to have fixed effects for prefecture but this seems impossible given that there is only one prefecture that is getting the treatment and it is unclear what omission of Hyogo prefecture means (this becomes the referent value?). Their defense relates to power (sufficient number of observations) which isn't the problem. Finally, the references cited in defense of this choice are not relevant to this particular paper. The current paper is a repeat cross-sectional design and the only treatment group is one prefecture and two of the three time periods. One reference is a within-person change study and the other looks at implementation of a federal policy and different states implemented at different times.

Reviewer 2 Report

Thank you for your revision.

Author Response

Thank you for your reviews and comments.